# Using Rotational Thromboelastometry to Identify Early Allograft Dysfunction after Living Donor Liver Transplantation

**DOI:** 10.3390/jcm10153401

**Published:** 2021-07-30

**Authors:** Chen-Fang Lee, Hao-Chien Hung, Wei-Chen Lee

**Affiliations:** 1Department of Liver and Transplantation Surgery, Chang-Gung Memorial Hospital at Linkou, Taoyuan City 333, Taiwan; chenfanglee@gmail.com (C.-F.L.); weichen@cgmh.org.tw (W.-C.L.); 2College of Medicine, Chang-Gung University, Taoyuan City 333, Taiwan

**Keywords:** coagulation, rotational thromboelastometry, early allograft function, liver transplantation

## Abstract

Background: Diagnostic tests for early allograft dysfunction (EAD) after living donor liver transplantation (LDLT) vary widely. We aimed to evaluate the predictive value of rotational thromboelastometry (ROTEM)-derived parameters in EAD. Materials and Methods: A total of 121 patients were reviewed. The definition of EAD proposed by Olthoff et al. included the presence of any of the following at postoperative day 7: bilirubin level ≥ 10 mg/dL, INR ≥ 1.6, or serum AST or ALT levels > 2000 IU/L. All patients underwent ROTEM assay, which consisted of an extrinsically activated thromboelastometric test (EXTEM) before and 24 h after LDLT. Results: The 1-year/2-year OS were 68.%8/64.5% and 94.4%/90.8% for the EAD and non-EAD groups, respectively (*p* = 0.001). Two independent risks were identified for EAD, the postoperative clotting time (CT, *p* = 0.026) and time to maximum clot firmness (maximum clot firmness (MCF)-t, *p* = 0.009) on the EXTEM. CT yielded a specificity of 82.0% and negative predictive value of 83.0%, and MCF-t displayed a specificity of 76.4% and negative predictive value of 81.9% in diagnosing EAD. The use of the 24 h post-LDLT ROTEM increased the effectiveness of predicting overall survival (OS) compared to using the Olthoff’s EAD criteria alone (*p* < 0.001). Conclusion: We conclude that CT and MCF on EXTEM were independent predictors of EAD. The 24 h post-LDLT ROTEM can be used with conventional laboratory tests to diagnose EAD. It increases the effectiveness of predicting OS.

## 1. Introduction

Early allograft dysfunction (EAD) is a major challenge in liver transplantation (LT), with an incidence rate ranging from 23.2% to 38.7% [1,2,3]. EAD indicates that the transplanted allograft does not work sufficiently and may lead to subsequent postoperative complications or even mortality [1,4]. Early identification of EAD is crucial for the implementation of immediate proper treatment [5]. However, the diagnostic tests of EAD vary widely [1,6,7], and there is a great disparity in the diagnostic period, ranging from 48 h to one week after transplantation. In addition, the definition of EAD uses traditional parameters such as lactate, bilirubin, transaminases level, and prothrombin time (PT) which can easily be altered by non-hepatic diseases. Therefore, a more reliable diagnostic tool is required.

Viscoelastic tests such as thromboelastography and rotational thromboelastometry (ROTEM) provide a quick and integrated evaluation of the coagulation function, thus reducing unnecessary blood transfusions. These blood tests provide a sophisticated conception of hemostatic pathways. Clinically, these studies have been used as early indicators to evaluate coagulopathy, either hyper- or hypo-coagulation. It has been proven that viscoelastic tests can identify bleeding tendency in patients who are to undergo major operations [8,9,10]. ROTEM provides advanced thromboelastography and permits a rapid and thorough evaluation using a small amount of whole blood [11]. ROTEM-derived parameters detect early signs of bleeding or liver dysfunction in both intrinsically and extrinsically activated assays in LT [12,13,14]. The real allograft function might not be able to be truly reflected by elevated liver function tests (transaminases), excessive bleeding after transplant, and the amount of needed blood transfusion, which were considered as possible signs of EAD [7,8,12]. In the present study, we aimed to evaluate the predictive value of post-transplant ROTEM-derived parameters on EAD and to validate their relationship with graft and patient survival.

## 2. Materials and Methods

### 2.1. Patient Enrollment

This retrospective and observational study was conducted at Chang Gung Memorial Hospital, Linkou branch, in Taiwan. We consecutively enrolled living donor liver transplants (LDLT) between January 2016 and December 2018. We excluded patients on any medication known to interfere with coagulation function, who had received blood transfusions within 48 h before LT or had a known hemostatic disease other than cirrhosis. Subsequently, we included 121 patients, who we further stratified into two groups (EAD and non-EAD). The hospital ethics committee approved this study (CGMH IRB 202001558B0).

### 2.2. Definition of EAD and Graft Failure

We adopted the definition of EAD proposed by Olthoff et al., which classifies EAD based on the presence of any of the following: hyperbilirubinemia (>10 mg/dL), prolonged international normalized ratio (INR > 1.6) at postoperative day 7, and/or aspartate aminotransferase (AST) or alanine aminotransferase (ALT) > 2000 IU/L within the first 7 days [2].

The graft failure was defined by post-transplant clinical manifestations of the recipient rather than death or re-transplantation: persistent hyperbilirubinemia > 20 mg/dL, continuing prolongation of INR > 3.0, refractory encephalopathy with progressive increase in serum transaminase level > 2500 IU/L.

### 2.3. Data Extraction and Clinical Outcome Measure

Basic demographics, routine laboratory tests including conventional coagulation tests, surgical information, and blood product transfusion records were documented. EAD-associated factors including AST, ALT, serum bilirubin, and international normalized ratio (INR) levels were checked at least once daily until postoperative day seven. Post-transplant clinical outcomes including length of stay in the intensive care unit (ICU), complications, graft survival, and overall survival (OS) rates were analyzed. Postoperative complications were recorded, and major complications were defined as a grade equal to or beyond Clavien–Dindo classification grade IIIa [15]. Hospital or surgical mortality was determined as death occurring within 30 days after transplant or during the same hospitalization period without discharge, irrespective of the length of stay (LOS).

### 2.4. ROTEM Assays

Rotational thromboelastometry (ROTEM^®^, TEM International GmbH, Munich, Germany) assessments were performed before and 24 h after transplantation in accordance with the manufacturer’s instructions. The ROTEM assays consisted of an extrinsically activated thromboelastometric test (EXTEM) and fibrin polymerization test (FIBTEM) to evaluate the extrinsic pathway, and the contribution of fibrinogen to clot formation. The ROTEM-derived parameters analyzed in our study included the clotting time (CT), clot formation time (CFT), maximum clot firmness (MCF), time to maximum clot firmness (MCF-t), lysis index at 30 min (LI30), clot formation rate (CFR), alpha angle, and maximum clot elasticity (MCE). The CT was defined as the period between the initiation of the ROTEM assay and a clot firmness amplitude of 2 mm.

### 2.5. Statistical Analysis

Clinical parameters with categorical values (expressed as numbers and percentages) were compared using Pearson’s chi-square test, while continuous variables (expressed as mean ± standard deviations) were analyzed using the independent *t*-test. Survival was compared using the Kaplan–Meier method. All potential risk factors for EAD were analyzed using the receiver operating characteristic (ROC) curve. The discrimination ability of EAD risk was examined using the area under curve (AUC), and an optimal cut-off value was identified using a Youden index, and presented along with its given sensitivity, specificity, positive predictive value (PPV), and negative predictive value (NPV). A two-tailed *p* value less than 0.05 was considered statistically significant. All calculations were performed using SPSS (SPSS Inc., Chicago, IL, USA).

## 3. Results

### 3.1. Patients’ Characteristics

The donors’ and recipients’ demographic background and surgical records in each group are shown in Table 1. We compared the data in the EAD (*n* = 32, 26.4%) and non-EAD (*n* = 89, 73.6%) groups and analyzed the associated risk factors. There were no proportional differences between the EAD and non-EAD groups regarding the donor/recipient age, body mass index, hepatitis status, and history of alcohol use. Although the incidence of hepatocellular carcinoma (43.8% vs. 25.0%, *p* = 0.061) and proportion of male sex (82.0% vs. 65.6%, *p* = 0.056) was higher in the non-EAD group than in the EAD group, the difference was not statistically significant. Regarding recipients’ conventional coagulation studies, aPTT level, fibrinogen concentration, or platelet count were not statistically significant. Notably, the PT (21.1 ± 5.7 vs. 18.3 ± 6.1, *p* = 0.012) significantly differed between the EAD and non-EAD groups. The mean model for end-stage liver disease (MELD) scores were 21.2 ± 10.4 and 16.3 ± 7.6, respectively (*p* = 0.019). With regard to surgery-related information, an association between intraoperative blood component transfusion and EAD was observed (RBC, *p* = 0.009; platelet, *p* = 0.011; cryoprecipitate, *p* = 0.049). These data suggest that although coagulopathy in patients with higher MELD scores had been corrected with more blood transfusions during operation, they still had a higher chance of EAD after LDLT.

### 3.2. Comparisons of Postoperative Outcomes between the EAD and Non-EAD Groups

Next, we evaluated whether the patients who met the EAD criteria used in our study had inferior outcomes. Table 2 details the postoperative outcomes including ICU and hospitalization LOS, complications, surgical mortalities, graft failure rates, and 12-month overall survival. Compared to the non-EAD group, the EAD group experienced an increased LOS in the ICU (mean: 31.0 vs 24.4 days, *p* = 0.036) and hospitalization (mean: 56.5 vs 43.9 days, *p* = 0.238), reflecting the necessity of critical care and close monitoring in cases of concern. In the EAD group, 16 patients (50.0%) experienced major complications, and nine patients died (28.1%). The causes of death with respect to hospital mortalities in the EAD group included three acute rejections, two massive bleeding episodes, two fulminant infections, one primary non-function, and one portal flow insufficiency due to portal vein thrombus. Figure 1 shows post-transplant survival according to the presence or absence of EAD, and the 1-year/2-year OS were 68.8%/64.5% and 94.4%/90.8% for the EAD and non-EAD groups, respectively (*p* = 0.001).

### 3.3. A Longer Postoperative CT and MCF-t on EXTEM Were Associated with a Greater Chance of EAD

We sought to investigate whether ROTEM assays could help increase the predictive power. The pre- and post-transplant ROTEM-derived parameters in EXTEM and FIBTEM assays are displayed in Table 3. Apart from MCF-t on EXTEM, the preoperative EXTEM and FIBTEM parameters did not significantly differ between the two groups, indicating that the baseline ROTEM data in both groups were very close. We found that a longer postoperative CT (*p* = 0.041) and MCF-t (*p* = 0.005) on EXTEM were associated with a greater chance of EAD events. Furthermore, we identified two independent risks for EAD on multivariate analyses (Table 4), the postoperative CT (HR (95% CI) = 1.014 (1.002–1.026); *p* = 0.026) and MCF-t (HR (95% CI) = 1.002 (1.000–1.003); *p* = 0.009) on EXTEM, both of which predicted post-transplant EAD. ROC curves for these two factors (Figure 2) revealed the following: postoperative CT (AUC (95% CI) = 0.654 (0.536–0.772); *p* = 0.010 (and MCF-t (AUC (95% CI) = 0.690 (0.584–0.797); *p* = 0.001) on EXTEM. These two independent risks were further evaluated using the performance indices in Table 5. CT on EXTEM yielded a sensitivity of 53.1%, specificity of 82.0%, PPV of 51.5%, and NPV of 83.0% at an optimal cut-off value of 87.5 s. The respective cut-off value for MCF-t on EXTEM was 1870 s, and it displayed a sensitivity of 53.1%, specificity of 76.4%, PPV of 44.7%, and NPV of 81.9% in diagnosing EAD.

### 3.4. Using 24 h Post-LDLT ROTEM Increased the Effectiveness of Predicting OS

Finally, we compared the cumulative OS for those meeting the Olthoff’s definition or 24 h post-LDLT ROTEM or not (either CT or MCF-t on EXTEM beyond the given cut-off values). We found that patients who met both had the most inferior results (Figure 3). Therefore, using the 24 h post-LDLT ROTEM increased the effectiveness of predicting OS after LDLT compared to adopting Olthoff’s EAD criteria alone (*p* < 0.001). These data encouraged us to use the 24 h post-LDLT ROTEM with conventional coagulation tests in diagnosing or excluding EAD.

## 4. Discussion

Our study demonstrated that postoperative EXTEM-derived parameters, including longer MCF-t and CT, were associated with a greater incidence of EAD in LDLT. It was observed that pre-transplant MELD scores also influenced the incidence of EAD. Early hemostatic dysfunctions could be detected by ROTEM studies as an increased CFT and decreased MCF in a previous study [13]. Similar to other studies, we found that massive blood loss and blood component transfusion were associated with the development of EAD; however, different ROTEM-derived factors were evaluated [13]. It appears that there is no general agreement about the EAD criteria, but a growing consensus has shown that a good EAD diagnosis should correlate with early graft failure and patient outcomes. Thus, we chose a previously deliberated and validated EAD definition, which revealed a strong connection between EAD and inferior outcomes.

The 24 h post-LDLT ROTEM coagulation times, including the CT and MCF-t, depend on the coagulation activation cascade and the concentration of coagulators [16]. It was evident that a prolonged CT on EXTEM correlates with higher PT levels, while MCF-t on EXTEM reflected fibrinogen activity and was related to platelet counts and intraoperative blood loss [17]. Meanwhile, systemic hypoperfusion may also be associated with prolonged coagulation times by affecting thrombin generation [18], thrombomodulin expression from ischemia-sensitive endothelia, altering hemostasis, and leading to hypocoagulability [19].

Another study reported similar fibrinolysis velocities between cirrhotic patients and healthy controls [20]. This could indirectly explain why the extrinsic pathway is more indicative of EAD than the intrinsic pathway preoperatively and at the early stage after transplant. In the present study, we used the post-transplant day 7 laboratory criteria to diagnose EAD, using liver function tests, the prothrombin time, and bilirubin level to determine the liver biosynthetic capacity [2]. Under certain definitions, it minimizes the eventuality of incomplete remission from preoperative hyperbilirubinemia and perioperative coagulopathy. Therefore, a clearly stated negative impact on both immediate post-transplant course and 2-year OS was observed in patients who met the EAD criteria.

The incidence of EAD in our study was 26.4%, which was close to that observed in previous studies that used the same criteria [2]. A greater proportion of recipients in the EAD group experienced graft failure (21.9%) compared to those in the non-EAD group (2.2%) (RR = 9.7, *p* < 0.001). Since 1998, many studies have focused on studying EAD and its associated risks, and their results varied based on their different definitions. The donor age, MELD score, PT and serum bilirubin levels prior to transplantation, acute deterioration, thrombocytopenia, blood loss and blood component transfusion, hepatic steatosis type and severity, and cold and warm ischemia times may all play a role in EAD occurrence [2,7,21,22,23,24,25,26,27,28,29,30]. The development of EAD is multifactorial, although many factors are inter-related. The MELD score incorporates the recipient serum bilirubin, PT, and creatinine levels prior to transplantation [31]. In this study, we only included LDLTs to minimize the impact of great differences in graft preservation time, which is a known major risk factor for EAD [21,32]. 

ROTEM assays simulate an in vivo situation of coagulation factors and thrombin generation [33]. Two independent coagulation pathways, extrinsic and intrinsic, were evaluated separately. Fibrinogen was converted to fibrin through a thrombin-dependent process. Complex mechanisms of intercellular communications and feedbacks, endothelium-released thrombomodulin, and natural anticoagulants influence particular stages of the clotting cascade [34]. The implementation of ROTEM in managing acute bleeding episodes has been widely studied and applied as a precursor in treating coagulopathy to lower bleeding-associated complications, morbidities, and mortalities [35]. Different ROTEM references have been objectively set up using various conceptual elements in healthy populations at all ages [36,37,38,39]. Additionally, unequal cut-off values in ROTEM parameters were also set to guide decision-making algorithms on the basis of differing backgrounds. In cirrhotic patients, ROTEM has a role in assessing the degree of liver dysfunction [40]. Conversely, the use of ROTEM in evaluating the coagulation status of cirrhotic patients may be less accurate because of hypoprothrombinemia and lack of sensitivity to thrombomodulin. Hypo-coagulation revealed by ROTEM assay correlates with an impaired liver function rather than an ongoing bleeding episode [41].

In the present study, there may be a divergence in the optimal cut-off value calculated by the Youden index from other institutes, depending on different laboratory settings. Although ROTEM displays more physiologic characteristics than conventional coagulation tests, the clinical implementation subjectively requires delicate interpretation [42]. As PT reflects thrombin generation initiated via the extrinsic pathway, there may be competition between PT and preoperative EXTEM-derived parameter. However, postoperative CT on EXTEM remained a significant predictor of EAD in the multivariate analysis. Besides, two independent risks identified in multivariate analysis, postoperative CT and MCF-t on EXTEM, demonstrated a good NPV (83.0% and 81.9%) on the optimal cut-off point. This means that a negative ROTEM screen test will help clinicians exclude the probability of EAD.

In addition to the study design (retrospective, single center), the main limitation of our study was that we focused on analyzing ROTEM-derived parameters rather than reading an objective morphology. There might be a practical gap between clinical research and application. Post-transplant blood product transfusion, coagulation tests, and ROTEM data responses were not taken into account for further analyses. Since there is no current consensus in blood product transfusion for post-transplant care and prospective and a large number of studies are required to draw valid effects of transfusion therapy, anti-coagulants on Rotem assays, our results should be interpreted with caution. From previous studies, two factors considered to be associated with EAD, acute deterioration and hepatic steatosis type and severity, were not available in our study.

In conclusion, our study demonstrates that the CT and maximum clot firmness on EXTEM-ROTEM performed 24 h after LDLT were independent predictors of early graft dysfunction, which in turn was associated with worse overall survival. To improve the overall results, ROTEM can be used as an auxiliary for prompt identification of EAD after LDLT.

## Figures and Tables

**Figure 1 jcm-10-03401-f001:**
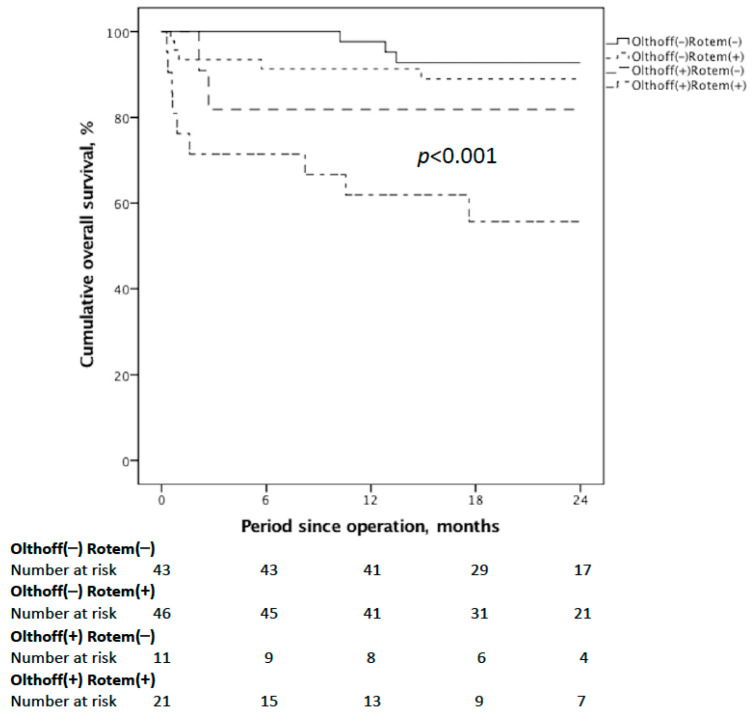
OS according to the development of EAD. Post-transplant 1-year/2-year OS were inferior in the EAD group than the non-EAD group (*p* = 0.001).

**Figure 2 jcm-10-03401-f002:**
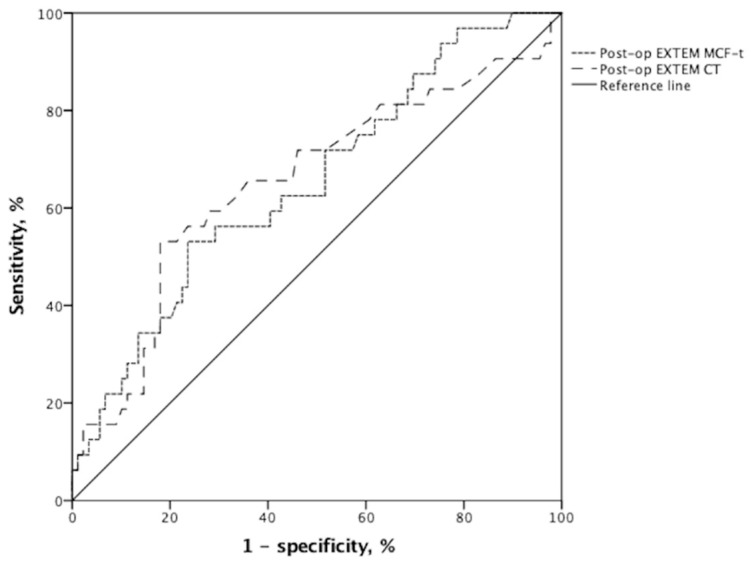
Receiver operating characteristic diagram for independent risk factors predicting EAD after LDLT: postoperative EXTEM CT (AUC (95% CI) = 0.654 (0.536–0.772); *p* = 0.010) and EXTEM MCF-t (AUC (95% CI) = 0.690 (0.584–0.797); *p* = 0.001).

**Figure 3 jcm-10-03401-f003:**
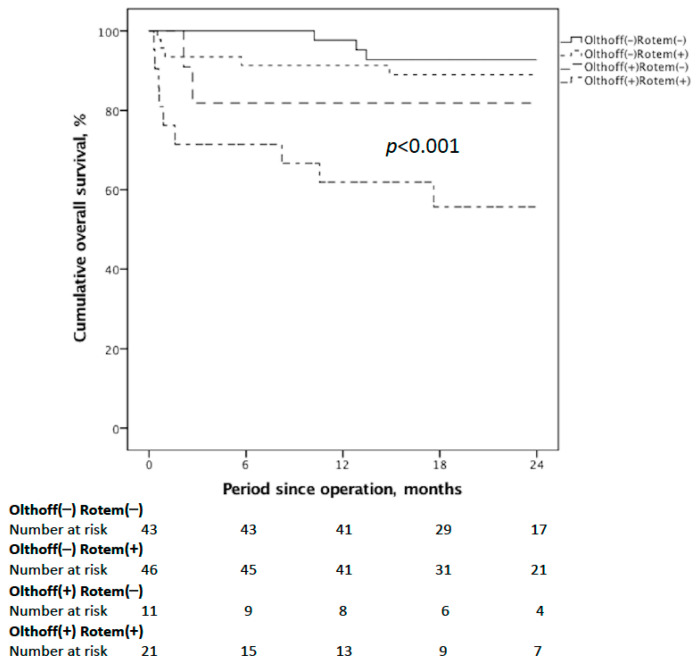
Cumulative OS curves for those meeting the Olthoff’s definition or 24 h post-LDLT ROTEM or not. ROTEM (+) is defined as at least one of the two above-mentioned 24 h post-LDLT ROTEM parameters being prolonged beyond the optimal cut-off values as mentioned. The Olthoff (+) ROTEM (+) group had the worst OS (*p* < 0.001).

**Table 1 jcm-10-03401-t001:** Demographics of patients after LDLT according to development of EAD or lack thereof.

KERRYPNX	EAD, N = 32	Non-EAD, N = 89	*p*-Value
**Basic conditions**			
**Recipient age, years old**	51.6 ± 9.9	53.6 ± 8.3	0.321
**Donor age, years old**	34.0 ± 8.7	31.3 ± 9.3	0.157
**Male gender, N (%)**	21 (65.6%)	73 (82.0%)	0.056
**Child-Pugh class C, N (%)**	12 (37.5%)	38 (42.7%)	0.595
**MELD**	21.2 ± 10.4	16.3 ± 7.6	0.019
**HCC, N (%)**	8 (25.0%)	39 (43.8%)	0.061
**HBV infection, N (%)**	12 (37.5%)	45 (50.6%)	0.204
**HCV infection, N (%)**	8 (25.0%)	15 (16.9%)	0.314
**Recipient BMI kg/m^2^**	25.4 ± 4.4	25.8 ± 4.4	0.683
**Donor BMI, kg/m^−2^**	23.5 ± 3.0	23.0 ± 3.3	0.493
**Alcohol use history, N (%)**	11 (42.3%)	35 (45.5%)	0.780
**PT, s**	21.1 ± 5.7	18.3 ± 6.1	0.012
**aPTT, s**	41.2 ± 12.0	39.1 ± 12.5	0.234
**Fibrinogen concentration, mg/dL^‒1^**	166.0 ± 93.7	159.3 ± 58.6	0.173
**Platelet count, nL^‒1^**	56.0 ± 20.0	60.2 ± 22.1	0.699
**Operation associated parameters**			
**Anhepatic time, mins** **Cold ischemia time, mins**	59.4 ± 46.6 67.1 ± 87.5	61.3 ± 39.0 56.3 ± 55.6	0.830 0.439
**Warm ischemia time, mins**	42.3 ± 42.7	40.2 ± 24.7	0.741
**GRWR, %** **Blood loss, mL**	0.96 ± 0.29 3270.3.8 ± 3421.4	0.97 ± 0.23 2081.8 ± 2321.7	0.825 0.076
**RBCs, units**	14.9 ± 15.0	7.3 ± 6.4	0.009
**FFP, units**	18.3 ± 14.3	13.6 ± 11.3	0.102
**Platelet, units** **Cryoprecipitate, units**	14.6 ± 12.5 4.1 ± 8.4	8.9 ± 10.2 2.3 ± 6.9	0.011 0.049

Abbreviations: MELD, Model for End-Stage Liver Disease; HCC, hepatocellular carcinoma; HBV, hepatitis B virus; HCV, hepatitis C virus; BMI, body mass index; PT, prothrombin time; aPTT, activated partial thromboplastin time; GRWR, graft recipient weight ratio; RBC, red blood cell; FFP, fresh frozen plasma.

**Table 2 jcm-10-03401-t002:** Post-operative outcomes of the patients after LDLT according to development of EAD or not.

	EAD, N = 32	Non-EAD, N = 89	*p*-Value
**ICU stay, days**	31.0 ± 24.4	20.9 ± 15.9	0.036
**Hospitalization, days**	56.5 ± 58.9	43.9 ± 48.6	0.238
**Major complication, N (%)**	16 (50.0%)	14 (15.7%)	<0.001
**Surgical/Hospital mortality, N (%)**	9 (28.1%)	3 (3.4%)	<0.001
**Graft failure, N (%)**	7 (21.9%)	2 (2.2%)	<0.001
**12-month OS rate, %**	68.8	94.4	0.001

Abbreviations: EAD, early allograft dysfunction; ICU, intense care unit; OS, overall survival.

**Table 3 jcm-10-03401-t003:** Results of thromboelastometric assays according to development of EAD or lack thereof.

	EAD, N = 32	Non-EAD, N = 89	*p*-Value
**Pre-op Parameters**			
**EXTEM CT, sec**	84.7 ± 41.7	74.7 ± 25.3	0.165
**EXTEM CFT, sec**	325.9 ± 311.4	266.2 ± 133.4	0.353
**EXTEM MCF, mm**	42.9 ± 10.4	41.8 ± 7.8	0.571
**EXTEM MCF-t**	2096.0 ± 426.3	1894.6 ± 318.2	0.036
**EXTEM alpha angle**	56.8 ± 18.1	55.1 ± 14.0	0.633
**EXTEM LI30**	99.9 ± 0.4	99.8 ± 0.3	0.822
**EXTEM CFR**	64.5 ± 13.4	63.0 ± 11.6	0.600
**EXTEM MCE**	80.7 ± 32.6	75.0 ± 23.9	0.359
**FIBTEM CT, sec**	119.6 ± 168.0	72.6 ± 20.3	0.185
**FIBTEM MCF, mm**	10.4 ± 4.6	10.1 ± 4.7	0.728
**FIBTEM MCF-t**	940.5 ± 526.9	824.0 ± 369.4	0.327
**FIBTEM alpha angle**	67.4 ± 8.1	68.9 ± 9.7	0.565
**FIBTEM LI30**	99.3 ± 2.0	98.3 ± 4.3	0.208
**FIBTEM CFR**	70.7 ± 5.4	71.0 ± 6.0	0.825
**FIBTEM, MCE**	11.5 ± 6.3	10.9 ± 5.4	0.625
**Post-op parameters**			
**EXTEM CT, sec**	98.7 ± 44.2	81.0 ± 28.5	0.041
**EXTEM A10, mm**	35.3 ± 10.7	35.8 ± 9.6	0.784
**EXTEM MCF, mm**	38.3 ± 8.6	40.0 ± 8.4	0.327
**EXTEM MCF-t, sec**	1891.0 ± 398.5	1681.1 ± 334.6	***0.005***
**EXTEM alpha angle, °**	47.9 ± 12.8	49.9 ± 11.7	0.407
**EXTEM LI30, %**	99.8 ± 0.5	99.9 ± 0.4	0.413
**EXTEM CFR, °**	57.5 ± 11.0	59.6 ± 10.0	0.312
**EXTEM MCE**	64.9 ± 24.0	70.2 ± 27.3	0.339
**FIBTEM CT, sec**	136.5 ± 169.3	86.5 ± 81.1	0.185
**FIBTEM A10, mm**	9.3 ± 4.4	9.9 ± 4.3	0.506
**FIBTEM MCF, mm**	7.4 ± 3.6	9.4 ± 10.4	0.292
**FIBTEM MCF-t, sec**	943.3 ± 494.3	878.1 ± 426.0	0.488
**FIBTEM alpha angle, °**	62.5 ± 7.9	58.8 ± 14.2	0.507
**FIBTEM LI30, %**	99.5 ± 1.7	98.6 ± 3.3	0.182
**FIBTEM CFR, °**	64.9 ± 14.1	69.3 ± 7.5	0.201
**FIBTEM, MCE**	8.0 ± 4.3	9.3 ± 4.5	0.177

Abbreviations: EAD, early allograft dysfunction; pre-op, pre-operative; post-op, post-operative; CT, clotting time; A10, the amplitude of the ROTEM tracings at 10 min; MCF, maximum clot firmness; MCF-t, time to reach the MCF; LI30, lysis index at 30 min; CFR, clot formation rate; MCE, maximum clot elasticity.

**Table 4 jcm-10-03401-t004:** Univariate and multivariate analyses of predictors for EAD after LDLT.

	Univariate			Multivariate		
	HR	95%CI	*p*-Value	HR	95%CI	*p*-Value
**EXTEM MCF-t, sec (pre-op) EXTEM CT, sec (post-op)**	1.001 1.014	1.000–1.002 1.002–1.025	0.035 0.017	1.014	1.002–1.026	0.026
**EXTEM MCF-t, sec (post-op)**	1.002	1.000–1.003	0.007	1.002	1.000–1.003	0.009

Abbreviations: EAD, early allograft dysfunction; op, operation; MCF-t, time to reach the maximum clot firmness; CT, clotting time. All parameters in Table 3 were calculated in univariate analysis, and only significant results (*p* < 0.100) are shown in this table and evaluated in multivariate analysis.

**Table 5 jcm-10-03401-t005:** Accuracy of postoperative day-1 thromboelastometric studies in predicting EAD events.

Parameter	AUC (95% CI); *p*-Value	Optimal Cut-Off Point	Sensitivity	Specificity	PPV	NPV
**EXTEM MCF-t, sec** **EXTEM CT, sec**	0.690 (0.584–0.797); *p* = 0.001 0.654 (0.536–0.772); *p* = 0.010	>1870.0 >87.5	53.1% 53.1%	76.4% 82.0%	44.7% 51.5%	81.9% 83.0%

Abbreviations: EAD, early allograft dysfunction; CT, clotting time; MCF-t, time to reach the maximum clot firmness; AUC, area under curve; CI, confidence interval; PPV, positive predictive value; NPV, negative predictive value.

## Data Availability

The data presented in this study are available on request from the corresponding author.

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
