# Peer review of "Using Rotational Thromboelastometry to Identify Early Allograft Dysfunction after Living Donor Liver Transplantation"

_jcm, 2021, doi:10.3390/jcm10153401_

Round 1
Reviewer 1 Report
By analyzing pre- and 24h postoperative ROTEMS of 121 patients that underwent LDLT in Taiwan, the authors aimed to identify additional diagnostic tools to identify EAD.
They divided their cohort in patients that developed EAD as defined by Olthoff et al. and those who did not. They then looked at ROTEM parameters in both groups and found that EXTEM-CT and EXTEM MCF-t were significantly prolonged in patients in the EAD group. They then calculated cut-off points for both values and compared cumulative survival after LDLT using the Olthoff and ROTEM criteria. When both criteria were present, survival was significantly inferior compared to patients that only meet one or none of the criteria.
The authors really tried to develop a new or additional diagnostic tool for the detection of EAD, nevertheless I am not sure, if the proposed study has the necessary power (only 32 patients in the event group). In addition, there are some other comments I want to share with the authors:
Page 2 line 44 could can (there are two verbs, please delete one)
Page 2 line 48 the transition is rough and I am not sure to what studies the authors refer by “these studies”. Before the intro was on ROTEM and coagulation testing. This sentence would benefit from rephrasing.
Page 2 line 77 Clavien Dindo IIIa? Please be precise with the definitions.
Page 3 line 93 where all your continuous data contributed normally? You analyzed your data with a t-test and did not use any non-parametric tests. With the number of enrolled patients this seems not correct. Please explain why you choose this statistical test? How did you check the distribution of data?
Page 3: Statistical analysis: univariate and multivariate analysis is missing in the statistics section. Did you choose EAD as dependent variable? How come that none of the factors in Table 2 were significant in a univariate analysis? The differences between both groups were pretty significant (p < 0.001 for most factors). This seems odd to me.
Page 3 line 110: conventional coagulation studies: please specify if these were done in the donor or recipient
“In the present study, we aimed to evaluate the predictive value of post-transplant ROTEM-derived parameters on EAD and to validate their relationship with longer-term graft and patient survival.” Please mention the same endpoints in the abstract and the text (there is nothing about survival in the abstract).
Also, you did only look at ROTEN derived parameters and short-term patient survival (1 and 2-year survival is not long term!). I did not see any calculation on these parameters or EAD and graft survival in this manuscript. Please only mention aims that are addressed in the manuscript.
Page 4 line 129: please include the numbers of LOS.
Page 4 line 131: In the EAD group 9 patients died (3 rejection, 3 bleeding, 2 infection, 1 PNF, 1 PV thrombus = 10 patients) But the numbers add up to 10…did you mix up the numbers? Otherwise please explain. Can you please give the numbers at risk for your Kaplan Meier graphs?
Figure 3. Using 24h-post-LDLT ROTEM increased the effectiveness of predicting OS: it would be very interesting how many patients were in each of those Olthoff-Rotem combinations. How many were + in both? With a sensitivity of roughly over 50% for the ROTEM parameters and a PPV of only around 50% (or even less) it’s like flipping a coin to identify patients at risk. Even though the MCF-t AND CT values reached statistical significance it is debatable if a HR of 1.002 for instance for EXTEM MCF-t, sec (post-op) is clinically relevant.
The authors themselves have mentioned that post-transplant blood product transfusion, coagulation tests, and ROTEM data responses were not taken into account for further analyses. In addition, hepatic steatosis type and severity were not analyzed in their study. Is there any change that they could get this additional but crucial information? As MELD score was the only factor that was different I those patients, what was the reason why some experienced EAD with high morbidity and mortality and the others did not? Di you perform a higher number of right grafts with complex reconstruction? Were there differences in anatomy or extension of resection? GRWR was similar between both groups, but this doesn’t mean that there were not differences in graft type. What do you think is underlying cause for the development of EAD in your cohort? In the early phase after LT, patients often get blood transfusions and other blood products or substances that can affect ROTEM parameters. Could this be a factor that limits this study? How many of those patients have gotten drugs or products that affect ROTEM? Could this be a major bias?
Reviewer 2 Report
The authors provide an innovative way to detect early allograft dysfunction (EAD) after LDLT, using ROTEM assay.
The main interest of this study is the precocious detection of EAD compared to usual criteria. Moreover, this study provides data on putcomes of EAD after LDLT in a large cohort of patients, something rarely described in the litterature.
However , the major limitation of the study is the lack of a practical implication of identifying a patient at risk of developing EAD using ROTEM since there is no change in patients' management.
There are some minor issues with the manuscript:
-is the ROTEM assay performed routinely in all patients receiving LT in your center at POD1? If not, this could lead to a selection bias.
-some definitions of outcomes are not clearly defined in the methods section: graft failure?
-methods and results sections: how are the variables expressed in the tables? (mean or median values?)
-results section: the ICU and hospital LOS seem very long, even in non-EAD patients=> any comment?
Round 2
Reviewer 1 Report
The authors addressed all raised concerns and revised the manuscript accordingly. I have no further comments.